# Modeling Habitat Suitability of Migratory Birds from Remote Sensing Images Using Convolutional Neural Networks

**DOI:** 10.3390/ani8050066

**Published:** 2018-04-26

**Authors:** Jin-He Su, Ying-Chao Piao, Ze Luo, Bao-Ping Yan

**Affiliations:** 1Computer Network Information Center, Chinese Academy of Sciences, Beijing 100190, China; pyc@cnic.cn (Y.-C.P.); luoze@cnic.cn (Z.L.); ybp@cnic.cn (B.-P.Y.); 2University of Chinese Academy of Sciences, Beijing 100190, China

**Keywords:** 1-D convolution, bar-head goose, convolutional neural network, DBIC, habitat preference

## Abstract

**Simple Summary:**

The understanding of the spatio-temporal distribution of the species habitats would facilitate wildlife resource management and conservation efforts. Existing methods have poor performance due to the limited availability of training samples. More recently, location-aware sensors have been widely used to track animal movements. The aim of the study was to generate suitability maps of bar-head geese using movement data coupled with environmental parameters, such as remote sensing images and temperature data. Therefore, we modified a deep convolutional neural network for the multi-scale inputs. The results indicate that the proposed method can identify the areas with the dense goose species around Qinghai Lake. In addition, this approach might also be interesting for implementation in other species with different niche factors or in areas where biological survey data are scarce.

**Abstract:**

With the application of various data acquisition devices, a large number of animal movement data can be used to label presence data in remote sensing images and predict species distribution. In this paper, a two-stage classification approach for combining movement data and moderate-resolution remote sensing images was proposed. First, we introduced a new density-based clustering method to identify stopovers from migratory birds’ movement data and generated classification samples based on the clustering result. We split the remote sensing images into 16 × 16 patches and labeled them as positive samples if they have overlap with stopovers. Second, a multi-convolution neural network model is proposed for extracting the features from temperature data and remote sensing images, respectively. Then a Support Vector Machines (SVM) model was used to combine the features together and predict classification results eventually. The experimental analysis was carried out on public Landsat 5 TM images and a GPS dataset was collected on 29 birds over three years. The results indicated that our proposed method outperforms the existing baseline methods and was able to achieve good performance in habitat suitability prediction.

## 1. Introduction

Human population growth and continuous change in the global climate have had a negative effect on wild animals’ important stopover habitats. Understanding potential animal habitats has become one of the central topics in ecology, natural resource management, and animal protection [1]. In recent years, habitat suitability models have become increasingly important for studying species distribution patterns. Species distribution model uses associated animal presence data to predict the presence probability in other specific areas or the same area at a different time. It has been widely used to analyze the relationship between species and environment variables, especially the distribution patterns under the influence of climate change [2], the expanding area of invasive species [3], or shrinking habitat of endangered species [4]. Several existing species distribution methods such as BIOCLIM [5], DOMAIN, and maximum entropy method (MaxEnt) [2] can model potential distributions with presence-only data along with environmental information for the whole study area. For presence/absence data, logistic regression methods (LR), Support Vector Machines (SVM), and artificial neural network (ANN) are the most commonly used statistical procedures [3,6,7]. Indices derived from remote-sensing data such as the Normalized Difference Vegetation Index (NDVI) have been used extensively in species distribution models [4,6,8,9,10,11].

There are two prominent limitations of traditional niche-based species distribution models. First, the spatial biases in existing occurrence data present limitations. The occurrence data are generally derived from large-scale field survey, herbarium and museum collections, public literature, and more. The data from old museum and herbarium collections usually only have country names or city names without latitude and longitude information. Therefore, only a very limited amount of valid data could be acquired by those methods and it is easy to include sampling biases especially for endangered species. With the widespread application of various advanced data acquisition equipment such as Argos, GPS, and wireless trackers, a large amount of single species presence data can be obtained. Second, a lack of spatially explicit predictor variables to fully capture habitat characteristics of species [12]. Satellite data can be widely available in different spatial, temporal, and spectral resolutions and can provide spatially refined information of landscape and hydrological characteristics. Remote sensing data were proven to not only have the capability to improve biodiversity and rarity assessments especially in predictive studies covering extensive and remote areas [4,13] but also can be a potential tool for reducing the overestimation of species richness by stacked species distribution models [11]. Niche factors direct and indirect environmental gradients in terms of satellite data only use several bands and may lead to low performance in complicated situations. Therefore, it became important to develop new approaches to combine GPS tracks and satellite data for predicting the habitat suitable for species.

In recent years, deep learning has been successfully used in many domains and deep convolutional nets have brought about breakthroughs in processing images, video, speech, and audio [14]. Deep convolutional net is a representation-learning method that can automatically learn internal feature representations with multiple levels from original images instead of empirical feature design. Several existing deep convolutional nets such as AlexNet [15], VGG-VD network [16], and GoogLeNet [17] have gained great success for ImageNet [18] classification or other visual recognition applications. Deep convolutional nets were also proven to be very efficient in remote sensing image classification [19,20,21,22,23] and object detection [24,25]. However, a traditional two-dimensional convolutional neural networks (CNN) mainly design for a single date image that takes the remote sensing image as the only input, lacks the ability to take other niche factors into consideration such as temperature and precipitation. In this work, we propose to use multi-convolutional neural networks (M-CNN) to automate extract discriminative features from the temperature time series and satellite data at the same time.

In this paper, we design a two-stage algorithm to predict the habitat suitability of migratory. At the first stage, we focus on generating classification samples from raw movement data by detecting all the stopovers for animal. A new density-based increment clustering (DBIC) method is introduced to find stopovers from migratory birds’ trajectories. The migratory bird usually makes an annual long-distance migration. It is more reasonable to get presence data from their stopovers instead of treating the collected data as presence samples. At the second stage, we separate the raw remote sensing image into small patches and introduce a multi-convolution neural network architecture to make it capable for incorporating temperature points and remote sensing images. A nonlinear SVM classifier based on radial basis function (RBF-SVM) is then trained to assign a category to an entire image patch.

## 2. Methodology

In this section, we explain in detail the basic operations of our algorithm. A block diagram of the overall system is shown in Figure 1. Firstly, we extracted stopovers of interest from GPS tracks by DBIC algorithm. Secondly, the Landsat images along with temperature were divided into positive/negative samples according to whether the image has overlap with any stopovers. Thirdly, the classified samples were used to train an M-CNN network. Finally, the representative features were extracted from the fully-connected layer of the trained M-CNN model and were used to model the habitat suitability of migratory bird by SVM model.

### 2.1. DBIC

The DBSCAN [26] algorithm is an outstanding representative of density-based algorithm for finding the high-density areas in spatial data. We have modified DBSCAN to enable it to cluster spatial-temporal data. The DBIC, which is based on DBSCAN, redefine the spatio-temporal density of a point by introducing the concept of global density and trajectory density. The global density can be calculated as the sum of the influence value of all data points, which belong to other trajectories as well as within its spatial neighborhood. The trajectory density is the totality influence of a data point within its temporal neighborhood in the same trajectory. The influence intensity between two points can be calculated using a mathematical function such as parabolic functions, square wave functions, or Gaussian functions.

**Definition** **1 (Trajectory).**
*A trajectory T is a time-ordered sequence of spatio-temporal sample points $T=p_{1}\to p_{2}\ldots \to p_{n}$, where $p_{i}=\left(loc_{i},t_{i}\right)$, including the geographic coordinate and time-stamp.*


The trajectories may have different lengths and sampling frequencies resulting from the different position acquisition devices.

**Definition** **2 (Influence function).**
*The influence function of a point
y is defined in terms of a Gaussian function.*
(1)$$f_{Gauss}^{y}\left(x\right)=f\left(x,y\right)=e^{-\frac{d{\left(x,y\right)}^{2}}{2\sigma^{2}}}$$
*where
d(x,y) denote the Euclidean distance function between point x and y. In principle, the influence function can be an arbitrary function.*


In our method, unlike simply counting the number of points in the neighborhood, we use the sum influence of those points to measure the importance of a point. As shown in Figure 2b,c, the point p1 and point p2 may have the same weight in ST-DBSCAN [27], but the point p1 would gain greater weight than point p2 in our method, which means it is more likely to be a stopover. Bar-headed geese are usually gregarious, and many types of geese live in large flocks. For example, a breeding colony often contains hundreds of pairs. Therefore, we assumed that they are more likely to stay in areas with higher GPS point densities which correspond to a higher global density in our algorithm.

The trajectory density of a point p=(loc,t)∈T is defined as:(2)$$f^{Traj}\left(p\right)=\underset{x_{i}\in N_{\xi_{t}}\left(p\right)}{\sum }f\left(x_{i},p\right)$$
where $N_{\xi_{t}}\left(p\right)=\{p_{i}|p_{i}\left(loc_{i},t_{i}\right)\in T,\;\left|t_{i}-t\right|\leq \xi_{t}\}$ is the subset of points that are temporal close to point p on the same trajectory (red points on the red line in Figure 2a).

The global density of a point p=(loc,t)∈T is defined as:(3)$$f^{Global}\left(p\right)=\underset{x_{i}\in N_{\xi_{d}}\left(p\right)}{\sum }f\left(x_{i},p\right)$$
where $N_{\xi_{d}}\left(p\right)=\{p_{i}|p_{i}\left(loc_{i},t_{i}\right)\notin T,\;d\left(p_{i},p\right)\leq \xi_{d}\}$ is the subset of points that are spatial close to point p on other trajectories (light blue points within blue shadow circle in Figure 2a).

The density of a point p can be denoted below.
(4)$$f\left(p\right)=f^{Traj}\left(p\right)+\alpha *f^{Global}\left(p\right)$$
where α∈[0,1] is the proportion coefficient.

### 2.2. Classification with M-CNN

The most significant advantage of CNN is that it offers an algorithmic means for extracting features directly from the raw pixel images. CNN has already been regarded as the most effective deep learning approach due to its remarkable performance on benchmark dataset such as ImageNet [28]. Classic convolutional neural networks [29] consist of alternatively stacked convolutional layers, normalization, pooling layers and a fully-connected layer. The convolutional layer performs a convolution of the input with a filter and produces an output called an activation map. Several filters can be used in a single convolutional layer and the nonlinear activation maps (rectifier, sigmoid, tanh, etc.) of each filter are stacked to form the output of this layer called the feature map, which is an input to the next layer. The pooling layers perform a down sampling operation along the spatial dimensions of feature maps via computing the maximum on a local region. It is conducive to mitigating overfitting risk by reducing the dimensions of feature vectors by offering invariance [30] and increasing the receptive field. The fully-connected layer is a regular multi-layer perceptron in which a neuron is connected to all neurons in the previous layer and the last fully-connected layer is for classification. In this paper, one-dimensional convolution is utilized to automatically extract the temperature sequence features and two-dimensional convolution is applied to extract the hierarchical remote-sensing image-related features.

#### 2.2.1. 1-D Convolution

As seen in Reference [31], 1-D CNNs have been successfully used for hyperspectral image pixel-level classification. In the 1-D convolution operation, the input data is convolved with 1-D kernels (the length of 1-D kernels is the size of the receptive field) and then move through the activation function to form the output data (feature vectors). Using the linear rectifier as an activation function, the value at position i on the jth feature vector in lth layer is given by the equation below.
(5)$$y_{i,j}^{l}=\max\left(\Sigma_{m}\Sigma_{k}w_{k}^{l}y_{i+k,m}^{l-1},0\right)$$
where *l* is the current layer number and *m* is the length of feature vectors in the previous layer connected to the current feature vector, k denotes the kernel size and $w_{k}^{l}$ the kth value of kernel.

#### 2.2.2. 2-D Convolution

Similar to 1-D convolution, the input data of the 2-D convolution is convolved with 2-D kernels and then go through the activation function to form the output data. Using the linear rectifier as an example, the feature map can be calculated by the following equation.
(6)$$y_{i,j,k}^{l}=\max\left(\Sigma_{m}\Sigma_{w,h}w_{w,h}^{l}y_{i+w,j+h,m}^{l-1},\;0\right)$$
where (i,j) is the pixel index in the feature map, $y_{i,j,k}^{l}$ stands for the value at location (i,j) in kth channel, (w,h) denote the width and height of 2-D kernel, and k is used to index the channels of the feature map.

#### 2.2.3. Network Architecture of Our Method

Deep neural networks require a lot of training data to learn deep structure and its related parameters. We can employ a pre-trained network and fine-tune the network on our new training images or feed the images to pre-trained CNNs for feature generation. Those models usually are highly correlated with their input data and have a very deep network. They also require a fixed-size (e.g., 224 × 224 if they pre-train on ImageNet) input image and produce a large dimension output features such as AlexNet [15], VGG-VD network [16], and GoogLeNet [17]. Beforehand, we need to resize each image scene to the fixed size by feeding it into the network. The size constraint causes inevitable degradation in spatial resolution when the original size of the image has too large a difference with the pre-defined size of the CNN. Therefore, we designed a simplified M-CNN which refers to common convolutional neural networks and comprises 1-D convolution and 2-D convolution at the same time. A schematic overview of the proposed CNN architecture is shown in Figure 3. After the M-CNN model is trained, the 512-dimensional feature vectors extracted from FC layer are used as input to train a non-linear SVM classifier. For the 2-D convolution layer, we referenced the architecture of Inception module in GoogLeNet [17]. The Inception module applies filters of different sizes at the same layer to maintain more spatial information and reduce the number of parameters of the network. As features of higher abstraction are captured by higher layers, the filter size could be bigger when move to higher layers. In our model, we fed into the network with a small image size of 16 ×16 pixels which means only a few filter sizes are available. Therefore, we apply three different sizes of filters to the raw input images and use 3 × 3 convolutions in the last two convolution layers.

## 3. Experiment

To evaluate the effectiveness of our proposed method, we performed a set of experiments and compared the results with several existing methods. The datasets and the details about the experiments conducted are presented in the following subsections.

### 3.1. Data

There are three types of data sources in our experiments. A real trajectory dataset from 29 bar-headed goose (BHG) used for clustering from 2009 to 2009 was collected from a satellite tracking project. The Landsat 5 TM images were acquired from U.S. Geological Survey (USGS) [32] and temperature data downloaded from national oceanic and atmospheric administration (NOOA) [33].

#### 3.1.1. Movement Data

The bar-headed goose (Figure 4) is one of long distant migrant birds in Asia that breeds in colonies of thousands near mountain lakes and winters in low-latitudes. They need suitable staging and stopover sites along their flight routes to complete their migration. They prefer to group activities in breeding season, wintering season and migration season and they also converge to stop-over, molt or breed in Qinghai Lake which is the largest saltwater lake in China.

We use a real GPS dataset of 29 bar-headed geese (BHG) tracked from March 2007 to January 2010 in Qinghai Lake National Nature Reserve, Qinghai province, China. BHG were captured and marked at three sites at Qinghai Lake including Jiangxigou, Hadatan, and Heimahe. BHG were captured on 25–31 March 2007 and 28 March–3 April 2008 using monofilament leg nooses (made by Indian trappers). Each bird was equipped with a 45-g solar-powered portable transmitter terminal (PTT: Micro-wave Telemetry PTT-100, Columbia, MD, USA). PTTs measured 57 mm × 30 mm × 20 mm and were attached dorsally between the wings with a harness system. The devices were designed to record locations every two hours (see Table 1). However, a significant number of samples in the datasets are missing due to the loss of satellite signals or unstable devices. For example, data were lost when animals stayed inside a dense forest or during network transmission errors. Therefore, the actual recording intervals vary from several hours to ten days. Outlier records are removed, and the small amount of missing values are estimated and considered. The processed data, which contains 60, 161 points (blue point in Figure 5), are then stored at a relation database for further processing.

Figure 5 shows the clustering result of DBIC on bar-headed geese’s tracks, which find 290 stopovers between Qinghai Lake and wintering area in Tibet such as Hala Lake, Qinghai Lake, Lhasa River, and Yarlung Zangbo River.

#### 3.1.2. Landsat 5 TM

For each stopover, we download the closest Landsat 5 TM images in time and have overlaps in spatial. Afterward, we split the images into 16 × 16 pixels patches and mark the patches as positive/negative samples whether or not they have overlaps with stopover. The images with cloud cover of more than 20% were removed and bands 1, 2, 3, 4, 5, and 7 of the Landsat 5 TM data were selected as the data sources.

#### 3.1.3. Temperature Data

We get the temperature data from 11 weather stations (red points in Figure 5) through NOOA [33]. Every weather station records the average temperature every day. For each remote sensing image, we take the temperature seven days before and after (including 15 days in total). We measure the collected time of remote sensing image of the nearest weather station and the temperature of the remote sensing image.

#### 3.1.4. Data Augmentation

Usually, deep CNNs perform well with sufficient training data. However, the datasets may be highly unbalanced and only the limited positive labeled samples are available, which may lead to over-fitting. To address those issues, we adopt a simple but effective data augmentation method to generate additional data without introducing extra labeling costs. We do this by rotating the original positive samples by 90° and 180° respectively. After the augmentation operation, the number of training samples can be increased by a factor of two.

### 3.2. Baseline Method

We compare our approach with SVM making use of gray-level co-occurrence matrix (GLCM) features [34,35], DenseNet [36], CNN and CNN + SVM. All the models were implemented using the same training as well as validating and testing datasets.

For the GLCM approach, the mean, correlation, contrast, energy (Angular Second Moment), homogeneity, and maximal probability were extracted from the GLCM because these have been proved effective in classification. We reduced the number to 64 gray levels in intensity of the image and selected eight pairs of different distance and direction to represent the spatial relationships of pixels. For each image, there will be 64 values to describe the image texture.

To assess the effectiveness of our designed network, we remove the 1-D convolution part in M-CNN structure as a new CNN structure. We also compare our method with DenseNet which obtain significant improvements over the state-of-the-art items on most of four highly competitive object recognition benchmark tasks [36]. Considering the smaller size of input image, we use a simplified version of DenseNet structure which only has 3 dense blocks, as is shown in Table 2. Only the Landsat images are fed into those two models during training and testing.

We employ the overall accuracy, F1 score, the area under the curve (AUC) of ROC [37], precision, and recall as the indicators to evaluate the quality of the competing algorithms. These indexes are calculated from a confusion matrix.

### 3.3. Experimental Setup

We conduct three groups of experiment. In the first scenario, we evaluate the effectiveness of proposed method on the bar-headed geese tracks. The total number of obtain samples was 27,714 images with 8696 positive and 19,018 negative from Section 2.1. Since the datasets are unbalanced, we applied data augmentation to the positive images by flipping horizontally or vertically and picked 6065 images randomly to expand the data set. Then we randomly divided the data set into three parts: training, validation, and testing, and conducted experiments. We repeated it four times with different division ratios (Table 3) and we computed the mean and standard deviation of each indicate for every method. For SVM + GLCM, CNN, and M-CNN, the training and testing samples were used. For M-CNN + SVM, the train set and validation were used to train M-CNN and SVM, respectively.

In the second scenario, we used the trained models to predict the potential habitat of bar-headed goose around Qinghai Lake in 22 February and 14 August. For bar-headed goose, Qinghai Lake is an important breeding and post-breeding place, but it is not a wintering ground [38]. The period of bar-headed goose stay at Qinghai Lake in the summertime could be divided into five phases [39], which are pre-nesting, nesting (include breeding), molt migration, molting, and pre-autumn migration. In August, most of the goose population was in the late breeding stage or the molting stage. At those stages, the goose is likely to care for young goose if they successfully hatched or store fuel for autumn migration. The bar-headed goose during the winter are located in tropical and subtropical regions in the Indian subcontinent and along the Yarlung Zangbo River, Lhasa River, Penbo River, and Niang River valleys in southern Tibet [38,40]. They still stay in the wintering area or start spring migration in February. Therefore, we choose each image of those two months as “highly suitable habitat” and “lowly suitable habitat”, respectively.

In the third scenario, we select four samples of positive class for visualization of M-CNN features. To intuitively understand how the M-CNN work, we recover the original image from feature maps of each layer with deconvolution method proposed in [41]. The reconstructed images loss more details increasingly along with deeper layers. The four samples were generated from one Landsat image around Qinghai Lake in 14 August. They locate in Egg Island, Luci Island, Sankuaishi Island and Buhahe Estuary, respectively. All four samples were assigned to positive in the second scenario by our M-CNN model.

All the algorithms used in these experiments are implemented in Python and are executed on a single machine with Intel(R) Xeon(R) CPU E5-2620 and 64 GB memory. The CNN model is implanted in Tensorflow-0.9.0 library [39]. Once the network is set up, the weights and biases are initialized using normalized initialization [42] and are learned by using variants of the gradient descent algorithm. The algorithm requires us to compute the derivative of a loss function with respect to the network parameters using the backpropagation algorithm. In the context of classification, the cross-entropy loss function is used in combination with the SoftMax classifier.

## 4. Results

We first depict the results of using six methods on classification and prediction tasks. Then, we visualize the extracted features from different convolutional layers by inverting feature maps into reconstruction images.

### 4.1. Classification Results

The classification results from GLCM + SVM, DenseNet, CNN, CNN + SVM, M-CNN, and M-CNN + SVM using a training sample size of images are compared in Table 4. There are three notable points. Initially, our approach obtains the better indicator values when compared with the other methods. M-CNN shows good performance in terms of AUC and recall, while M-CNN + SVM achieves best performance in terms of accuracy, F1 score, and precision. In addition, using the same network structure, the transferred CNN feature-based method has higher classification accuracy than CNN. Lastly, DenseNet and GLCM achieve similar accuracy levels and the CNN approach performs better than those two. The classification accuracy of DenseNet is 0.742 while the classification accuracy of simplified CNN is 0.801.

### 4.2. Prediction Results

The potential habitat mapping from Landsat TM images of Qinghai Lake in August 2010 and February 2011 were displayed in Figure 6 and Figure 7, respectively. From these two figures, several important phenomena need to be taken care of. First, only a few discrete small areas were marked as “very high” in the GLCM + SVM model and CNN model (Figure 6a,c) and those areas along the boundary of lakes were marked as “very high” in other methods. Second, the surfaces of the lakes were basically labeled as “very low” in our proposed method (see Figure 6f) such as Gahai Lake, Shadao Lake, and Jinsha estuary. This is the most significant difference between our method and other methods. Third, in February (see Figure 7), all the models except SVM + GLCM and DenseNet show broadly similar spatial patterns. They classified most parts of the regions as “very low” for bar-headed goose, which means it is inappropriate for habitat.

### 4.3. Visualization of Feature Maps

A feature map is generated when a filter with learned weights is applied to the input image or the previous output data. To increase the representational power of neural networks, several filters would be used in each layer. Therefore, each layer has lots of feature maps and we pick a high activation one to visualize. Reconstructed images of each convolution layers and max pool layers from our trained M-CNN model are shown in Figure 8. The rows indicate the four different samples. The raw images are shown in first column and the other columns indicate different layers. Generally, the lower layers of the CNN in charge of detecting low level features while the higher layers detect more abstract features related to the semantic classes. In our experiment, corners and edges are more prominent in the layer Conv2, such as edges between island and lake in image 1–3, and the boundary of the Buhahe in image 4. The layer MaxPool1 can clearly get the target area and the layer Conv3 seems to try to smooth the boundary. The layer MaxPool2 shows entire areas with gradient boundary. Indeed, in the final layer, we observe that high values correspond to areas where the network detects the presence of bar-headed goose, such as Sankusishi Island in image 3 and wetlands on the either side of Buhahe in image 4, while low activations correspond to the unsuitable areas, such as lake.

## 5. Discussion

In the first scenario, our model outperforms several methods in terms of all indicators. Both the SVM + GLCM and DenseNet have poor performance in classification and prediction experiment. One possible explanation for SVM + GLCM is that GLCM cannot obtain effective textures from the small size of the input image. The network structure of DenseNet in our paper is similar to the original paper [36]. The poor performance problem may result from the model input that we feed into a smaller size of the image. This also indirectly shows that it is difficult to use a pre-trained deep neural network to extract discriminative features from our images. When we compared two CNN models, three indicators increased significantly for the CNNs + SVM model, which confirms that a post-processing step of SVM classification is still necessary for achieving a good performance when using a convolutional neural network approach.

In August, the goose disperse and wander in wetlands and estuaries especially in the northwest of Qinghai Lake [43]. Figure 9 shows the distribution of bar-headed goose around Qinghai lake area. We can see that the bar-headed geese are more concentrated in the northwest of the Qinghai Lake with less distribution in the southeast. The result of our model (see Figure 6f), which the “very high” area does not contain the southeast and northeast regions of Qinghai Lake is most consistent with Figure 9. The DenseNet, CNN + SVM, and M-CNN seem to overestimate the “very high” area. In addition, the potential habitat locating on the surface of the lake seems to be uncommon. Our approach also achieves better results than other baselines from this point of view. In February, several methods show good performance. One possibility is that our model learned well from the classification sample. The other possibility is that the snow cover or ice cover led to the great change of the remote sensing image and the Landsat images, which used to generate train samples, do not contain the snow cover or ice cover situation. We are inclined with the first possibility because of the poor performance of GLCM (see Figure 6a) and DenseNet (see Figure 6b).

Remote sensing data play an important role in the potential habitat prediction due to its frequently and readily available in different spatial, temporal, and spectral resolutions. By incorporating low sample rate satellite tracking data and remote sensing data in this case study, we have presented an approach for extracting stopovers and predicting potential habitat based on presence-only occurrence data for the bar-headed goose. As the 1-D convolution and 2-D convolution used here are eligible to describe one dimensional time series data and characterize multiple land surface dynamics based on Landsat images and can easily be adapted to other input data or study extents, the proposed approach offers good opportunities for species transferability.

## 6. Conclusions

In this paper, we introduced a new clustering approach for generating classification samples and designed a multi-convolutional neural network for modeling habitat suitability of migratory birds around Qinghai Lake using with GPS tracks and remote-sensing images. Our experiments showed that classification of small patch images using SVM relying on GLCM features or DenseNet have a low accuracy and poor quality. The M-CNN combined with a RBF-SVM classifier achieves the best performance in the prediction experiment. The approach outlined in this paper can be replicated to map habitat in other species, which has movement data. Future work may involve developing a method that better deals with high cloud cover Landsat images, which could further add my classification samples and model accuracy.

## Figures and Tables

**Figure 1 animals-08-00066-f001:**
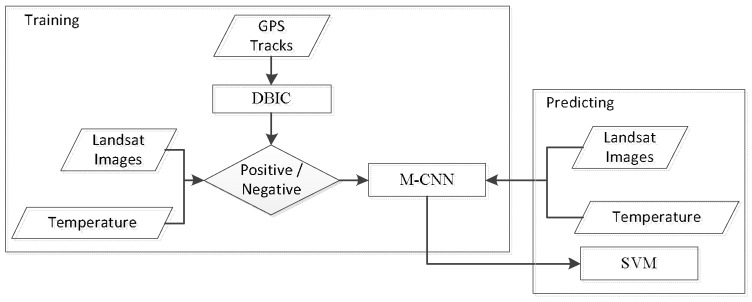
Block diagram of the prediction system. The parallelogram denote data, rectangle denote model and diamond denote discriminant operation.

**Figure 2 animals-08-00066-f002:**
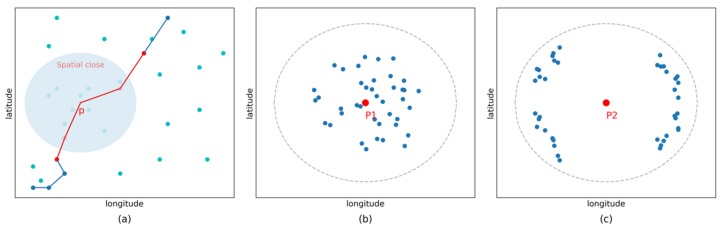
(**a**) Examples of spatial neighbor (light blue points within blue shadow circle) and temporal neighbor (red points on the red line). (**b**,**c**) Examples of core point with great different density.

**Figure 3 animals-08-00066-f003:**
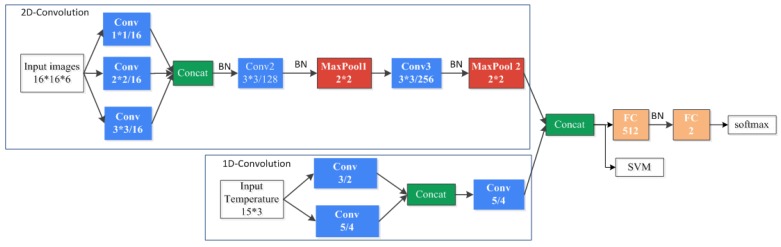
Overview of multi-convolutional neural networks (M-CNN) architecture. Note: conv denotes convolutional, batch normalization (BN) denotes batch normalization, FC denotes fully connected.

**Figure 4 animals-08-00066-f004:**
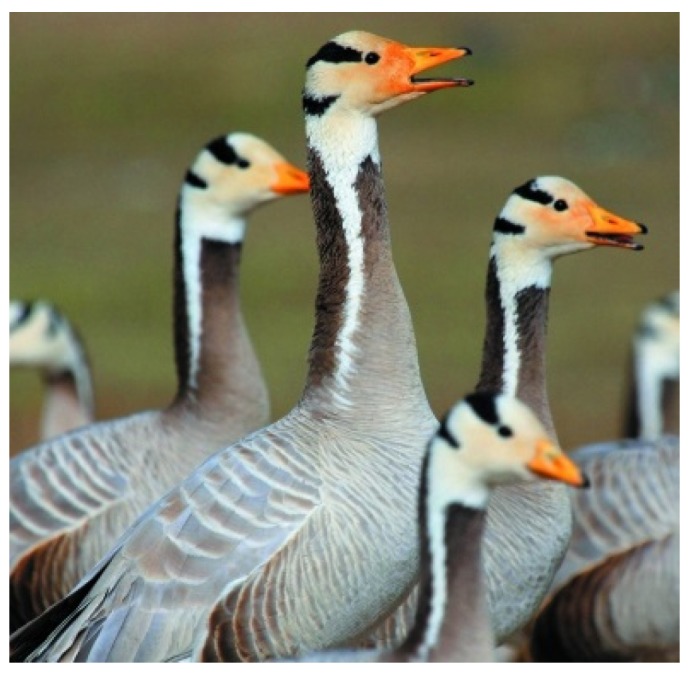
Examples of bar-headed goose.

**Figure 5 animals-08-00066-f005:**
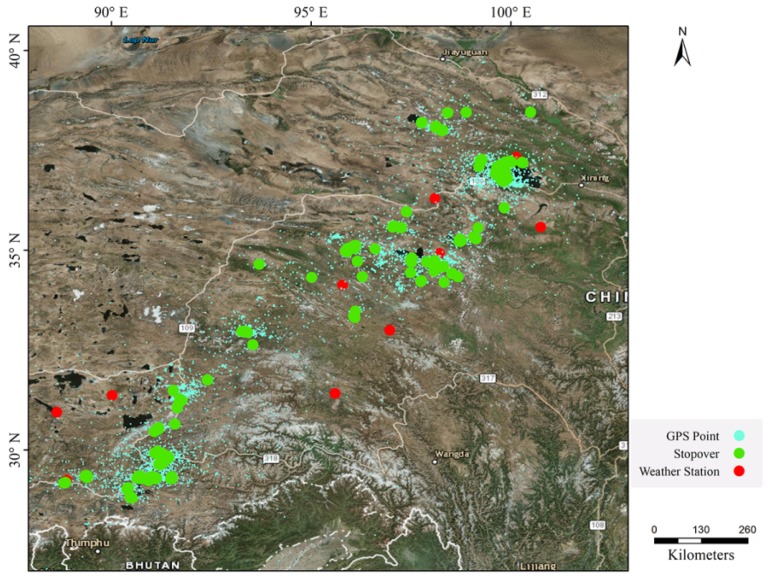
The distribution of stopovers. Blue points denote the original GPS point, green points denote the clustering result of DBIC, and red points denote the weather station.

**Figure 6 animals-08-00066-f006:**
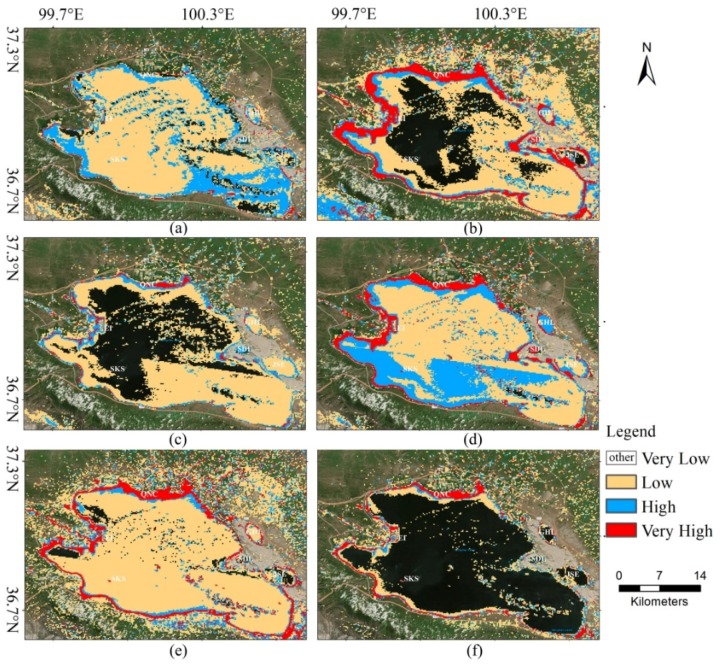
Predicted potential habitat of bar-headed goose around Qinghai Lake in August 2010. (**a**) SVM + GLCM, (**b**) DenseNet, (**c**) CNN, (**d**) CNN + SVM, (**e**) M-CNN, (**f**) M-CNN + SVM. Code: EI, Egg Island; SKS, Sankuaishi; QNC, Qinghaihu NongChang; GHL, Gahai Lake; SDL, Shadao Lake; JSE, Jinsha Estuary.

**Figure 7 animals-08-00066-f007:**
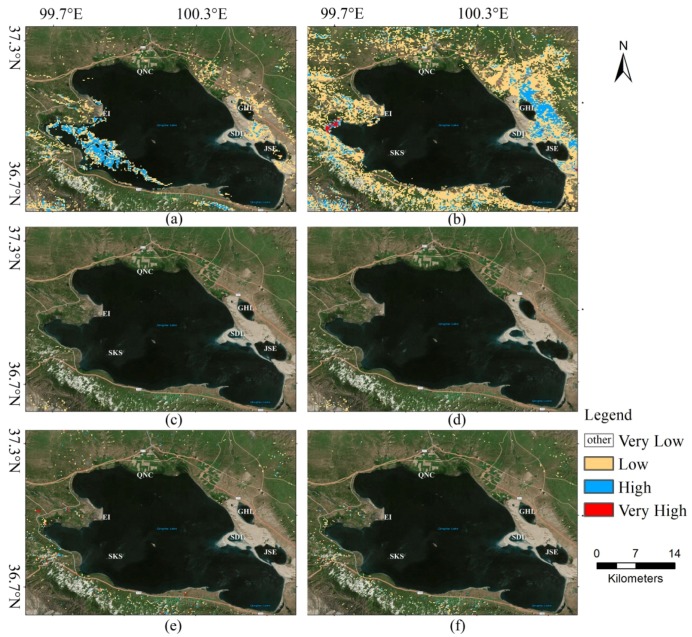
Predicted potential habitat of bar-headed goose around Qinghai Lake in February 2011. (**a**) SVM + GLCM, (**b**) DenseNet, (**c**) CNN, (**d**) CNN + SVM, (**e**) M-CNN, (**f**) M-CNN + SVM.

**Figure 8 animals-08-00066-f008:**
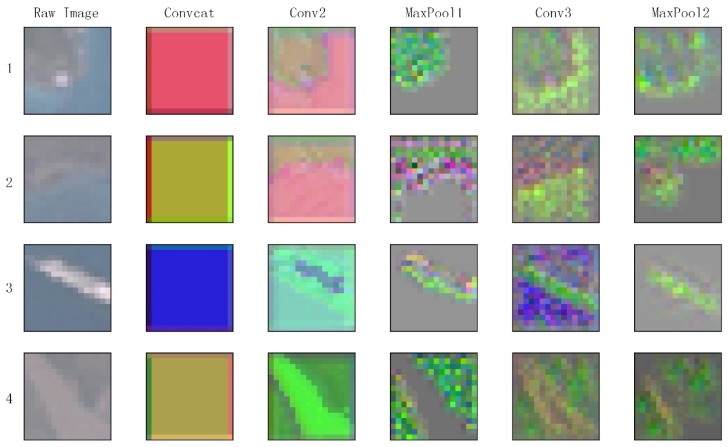
Recovered images of feature maps at different layers, derived from M-CNN model with four positive samples which locate in Egg Island, Luci Island, Sankuaishi Island and Buhahe Estuary, respectively.

**Figure 9 animals-08-00066-f009:**
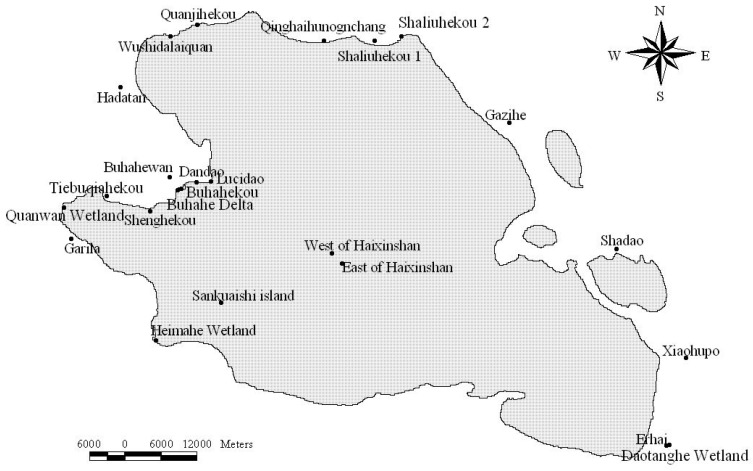
The distribution of bar-headed goose around Qinghai lake area [44].

**Table 1 animals-08-00066-t001:** Lon and lat denote longitude and latitude, and animal denote the identifier of animal.

Lat	Lon	Animal	Time
36.132	98.805	67,580	23 June 2007 15:16:04
36.609	99.19	67,580	23 July 2007 10:21:46
99.782	36.935	67,695	1 October 2007 5:00:00

**Table 2 animals-08-00066-t002:** A DenseNet with 3 dense blocks.

Layers	DenseNet	Output Size
Convolution	7 × 7/48	16 × 16
Max Pool	2 × 2	8 × 8
Dense Block 1	$\left[\begin{array}{c} 1\times 1\\ 3\times 3 \end{array}\right]\times 6$	8 × 8
Transition Layer 1	1 × 1 conv	8 × 8
	2 × 2 average pool	4 × 4
Dense Block2	$\left[\begin{array}{c} 1\times 1\\ 3\times 3 \end{array}\right]\times 12$	4 × 4
Transition Layer 2	1 × 1 conv	4 × 4
	2 × 2 average pool	2 × 2
Dense Block 3	$\left[\begin{array}{c} 1\times 1\\ 3\times 3 \end{array}\right]\times 16$	2 × 2
Classification Layer	global average pool	1 × 1
	SoftMax	1

**Table 3 animals-08-00066-t003:** Different data divisions for four experiments.

Training%	Validation%	Testing%
70	5	25
70	10	20
70	15	15
70	20	10

**Table 4 animals-08-00066-t004:** Comparison between approaches using GLCM + SVM, CNN, M-CNN, and M-CNN + SVM.

Method	Accuracy	F1	AUC	Precision	Recall
GLCM + SVM	0.769 ± 0.004	0.731 ± 0.005	0.839 ± 0.004	0.742 ± 0.005	0.719 ± 0.006
DenseNet	0.781 ± 0.023	0.768 ± 0.008	0.870 ± 0.008	0.713 ± 0.045	0.840 ± 0.060
CNN	0.803 ± 0.004	0.758 ± 0.018	0.880 ± 0.013	0.814 ± 0.038	0.715 ± 0.066
CNN + SVM	0.817 ± 0.008	0.780 ± 0.010	0.879 ± 0.011	0.817 ± 0.018	0.746 ± 0.016
M-CNN	0.835 ± 0.019	0.830 ± 0.021	**0.936 ± 0.020**	0.746 ± 0.027	**0.938 ± 0.041**
M-CNN + SVM	**0.864 ± 0.022**	**0.842 ± 0.029**	0.928 ± 0.015	**0.852 ± 0.017**	0.832 ± 0.044

Best scores are in bold.

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
