# Peer review of "Modeling Habitat Suitability of Migratory Birds from Remote Sensing Images Using Convolutional Neural Networks"

_animals, 2018, doi:10.3390/ani8050066_

Round 1

Reviewer 1 Report

This paper presents a multi-CNN framework for modelling habitat suitability of migratory bird from remote sensing images and temperature data. This topic is very interesting, seems to be a novel and suitable application of deep learning techniques. Good results have been also demonstrated via the proposed experimental results.

It is suggested to analyze or visualize the extracted features from different convolutional layers with their contributions to the final results.

It is suggested to display some training images and testing images, as well as discuss some failure cases.

It is suggested to discuss some related issues about selecting the number of layers or any concerns for designing this network.

It is suggested to provide some introductions with images to introduce the "Bar-head goose" considered in this paper.

Author Response

Reviewer 2

1.       It is suggested to analyze or visualize the extracted features from different convolutional layers with their contributions to the final results.

Response: we tried to explain how the M-CNN work by inverting feature maps into reconstruction images in section 4.3.

2.       It is suggested to display some training images and testing images, as well as discuss some failure cases.

Response: I am sorry. I don’t quite understand what I need to do exactly. In fact, the training images and testing images are actually divided from the same data set. Does the failure cases mean the negative samples? Should I display some image patches in my manuscript? I am not sure if it can meet you requirement that I have displayed some positive examples in section 4.3.

3.       It is suggested to discuss some related issues about selecting the number of layers or any concerns for designing this network.

Response: we have added introduction in section 2.2.3.

4.       It is suggested to provide some introductions with images to introduce the "Bar-head goose" considered in this paper.

Response: We brief introduced Bar-head goose and added an image in section 3.1.1.

Reviewer 2 Report

The study prepared a special dataset of bird presence and conducted supervised classification. The use of the GPS data provides a unique chance to delineate the spatial distribution of habitats. The classifier developed and selected in the study seems working, but the classification method section is not satisfactory.

The architecture of the deep network is not clear. For example, the number of channels/filters in each convolutional layer is not given. It is unclear how each of the models (for example, DenseNet) is implemented. As for the selected model, 1D convolution was applied to the temporal dimension and 2D convolution was applied to spatial dimensions. Why not just using 3D convolution? What is the logic of the design?

The study generally treats deep networks as a black box without trying to explain why they work. What factors motivated the authors to use CNN for habitat classification? Are there any temporal and spatial patterns recognized by the convolutional kernels? What is the seasonal dynamics and transitions of habitats? If habitats have unique spatial patterns, why the convolution size was set to 16*16? Without addressing these issues, the contribution to the research community is limited. The authors might want to consider visualization techniques for deep neural networks to demonstrate the usefulness of the customized network architecture.

Line 12: “existing methods”

Line 243: need a few references on GLCM-based classification.

Figure 5: the legend for “very low” seems wrong.

Author Response

Reviewer 1

1.       The architecture of the deep network is not clear. For example, the number of channels/filters in each convolutional layer is not given.

Response: We have redrawn the M-CNN architecture (see Figure 3) and added the channels/filters of each layer.

2.       It is unclear how each of the models (for example, DenseNet) is implemented.

Response: We have given the architecture of DenseNet in manuscript (see Table 2). The CNN model used in our manuscript is exactly the same as the M-CNN removed the 1-D convolution part.

3.       As for the selected model, 1D convolution was applied to the temporal dimension and 2D convolution was applied to spatial dimensions. Why not just using 3D convolution? What is the logic of the design?

Response: 3D CNN mainly used to extracts features from both spatial and temporal dimensions, such as for multi-spectral multi-temporal remote sensing data. In our case, 2D convolution is used to extract features from Landsat image at a single time point which corresponds to the time of bar-headed goose appeared.

How to choose the habitat of bar-headed geese is influenced by temperature, so we use 1D convolution in temperature data. And the 1D convolution could be added in other case if more than one environmental condition were included.

4.       The study generally treats deep networks as a black box without trying to explain why they work. What factors motivated the authors to use CNN for habitat classification? Are there any temporal and spatial patterns recognized by the convolutional kernels?

Response: we tried to explain how the M-CNN work in section 4.3.

5.       What is the seasonal dynamics and transitions of habitats?

Response: In our paper, we tended to propose a new method to model habitat suitability of migratory bird and didn’t focus on the pattern mining of habitats.

6.       If habitats have unique spatial patterns, why the convolution size was set to 16*16?

Response: There are two reasons to set the image size to 16*16. First, the size of image which is used to fed into cnn network is normally a power of two. Second, the resolution of Landsat image is 30m and the max positioning error of our PTT device is close to 500m.

7.       Line 12: “existing methods”

Response: We have revised this part according to your comment.

8.       Line 243: need a few references on GLCM-based classification.

Response: We have revised this part according to your comment.

9.       Figure 5: the legend for “very low” seems wrong.

Response: We have redrawn the legend of Figure 6 and Figure 7.

Author Response

Reviewer 3

1.       Line 34: Please arrange the keywords in an alphabetical order.

Response: We have revised this part according to your comment.

2.       Line 37: Please rewrite this sentence and make sure that it is coherent with the next sentence.

Response: We have revised this part according to your comment.

3.       Line 46: need a space in BIOCLIM[5]

Response: We have revised this part according to your comment.

4.       Line 91: 2. Methodology

Need more text here. Please explain why you choose this methodology.

Response: We have revised this part in 2.2.3.

5.       Line 95. Figure 1. Please explain the block presented in Figure 1, so that a general reader can understand the workflow of your methodology.

Response: We added brief description in line 95-101.

6.       Line 111: Please rewrite this sentence.

Response: We have rewritten this sentence.

7.       Line 114: The font inside the equations and the font inside the text is different. Please check this irregularity throughout the paper.

Response: I am sorry for that. I got a problem when I tried to revise my manuscript according to this comment. I typed the equations with the Microsoft Equation Editor. If I want to change the font styles, I need to change the format of equations from “professional type” to “plain text”. After done that, my Microsoft Word did not work properly and it makes me very difficult to continue to revise my manuscript. I would really appreciate that I could change the font styles of equations if my paper is lucky enough to be further processed.

8.       Line 152-154: ------the last fully-connected layer is a Softmax layer that-------- This statement is not necessarily true for all cases.

Response: We have rewritten this sentence.

9.       Line 187: Please redraw Figure 3.

Response: We have redrawn Figure 3.

10.   Line 192: 3. Experiment

When you start a section or subsection, at least you need to provide some text to help the reader. Please check this throughout your writing.

Response: We have added brief summaries at the beginning of each section/subsection.

11.   Line 205: Please put the reference in the reference section not in the main text.

Line 221: Please put the reference in the reference section not in the main text.

Line 226: Please put the reference in the reference section not in the main text.

Response: We have revised this part.

12.   Line 238: You did not define what GLCM is. Please define it.

Response: We have revised this part.

13.   A few comments about data preparation

a. You have done your experiment based on 90 degree and 180 degree augmentation. Along with this two degrees, pleases perform the experiment based on the augmentation using 45 and 135 degree.

Response: The bigger size of image patch, the lower spatial resolution of the final result. Hence, considering the spatial resolution, we split the Landsat images into relatively small image patches (16*16). The size would be changed if rotate an image 45/135 degree. We can crop picture and padding with zero values, but it would cause data loss which greatly increase the difficulty of model training.

14.   b. Please also provide your result without augmentation.

Response: we have provided a result without augmentation in the attachment. The result shows that data augmentation does not improve the accuracy conspicuously under the same data distribution (70-15-15). We could infer that the other experiments with different data distributions would achieve similar results.

In fact, the task in our manuscript just a 2-class problem and it has enough data to train the network. We apply data augmentation in our paper due to the following reason. In species distribution model, it is often easier to obtain presence data (positive sample). For many situations, the absence data are rarely available or it is difficult to define an area that absolutely unsuitable for species. Similarly, we can say that the stop overs found by DBIC (the clustering algorithm used in our manuscript) are suitable for bar-headed goose and could be used as presence data, but we can't guarantee that other places are unsuitable for bar-headed goose. This means that there are some error samples in negative samples. We try to increase the number of correct samples by rotating positive samples and the ratio of error samples is reduced automatically, which makes the model more robust.

15.   2. You have split your data in to 70% as Training, 20% as Validation and 10% as Testing. It is very difficult to comment on result based on this data distribution. I will suggest you to perform the whole experiment based on the following data distribution and compare the results.

Response: The detail results of each experiment were included in the attachment. In the manuscript, we changed to show the mean and standard deviation of each indicate for every method.

Round 2

Reviewer 2 Report

The revised manuscript has addressed all issues in the review report.

Reviewer 3 Report

Corrections have been done resonablre